# Post-Renewal Evaluation of an Urbanized Village with Cultural Resources Based on Multi Public Satisfaction: A Case Study of Nantou Ancient City in Shenzhen

Siming Gu, Jinqi Li, Mohan Wang and Hang Ma *

Shenzhen Graduate School, Harbin Institute of Technology, Shenzhen 518050, China; gsm1106@126.com (S.G.); lijinqi@hit.edu.cn (J.L.); wmhgogogo@163.com (M.W.)
* Correspondence: mahang@hit.edu.cn; Tel.: +86-13510018409

**Abstract:** The urban renewal of older districts usually has a significant impact on the sense of place and identity. However, a systematic post-renewal evaluation of older districts with cultural resources based on multi public satisfaction is still lacking. This study takes Nantou Ancient City in Shenzhen, an urbanized village with rich historical and cultural resources, as an example; this study introduces the multi public, including the residents, merchants and tourists, as the evaluation subject. By establishing an evaluation system suitable for the characteristics of Nantou Ancient City, this study explores the existing problems of renewal and transformation, summarizes experiences and lessons and provides a basis for the sustainable development of Nantou Ancient City and the organic renewal of urbanized villages in Shenzhen. First of all, on the basis of combing the existing community renewal evaluation system, especially the satisfaction evaluation and the research on the historical value of the urbanized villages, the historical and cultural value and the renewal and transformation process of Nantou Ancient City are introduced, and the multiple values of Nantou Ancient City are clarified. Secondly, the demand contradiction between the public group and the core stakeholders is analyzed, as well as the internal demand and difference of the public group. After that, combined with field research, literature analysis and network review data analysis, the post-renovation evaluation system of Nantou Ancient City is constructed in terms of six aspects: residential environment, supporting facilities, street space, history and culture, economy, commerce and social culture. Then, the questionnaire is designed for three different groups of residents, merchants and tourists, and the evaluation system is applied to Nantou Ancient City for empirical research. By means of mean analysis, variance analysis and IPA analysis, the similarities and differences of the evaluation of different public groups are compared, which is taken as the basis for summarizing the existing problems in the renovation of Nantou Ancient City, and optimization suggestions are put forward.

**Keywords:** urbanized village; Nantou Ancient City; public satisfaction; post-renewal evaluation



## 1. Introduction

With the reform and opening-up in 1978, China changed from a planned economy to a market economy; land ownership reform and the change and rapid development of China's urbanization led to academic research in many policies and related fields [1,2]. In the field of urban construction, high-speed economic development has gathered social wealth, and meanwhile, in order to support various economic activities, there is also a greater and diverse demand for the use and development of urban land. The expansion of urban land has led to the blurring of urban and rural boundaries, while the large number of job opportunities that cities can provide has led to the migration of people from rural to urban areas, and the demand for urban residential space is increasing. The urbanized village is a special type of city generated under the above historical development background of

China [3]. Urbanized villages first appeared in the coastal areas of southern China, such as Shenzhen and Xiamen in the 1980s, and then expanded to inland cities in the 1990s [4,5]. Although the urbanized villages are close to or appear in the city, they are usually full of low-quality informal housing lacking planning control [5,6].

After decades of rapid urbanization after reform and opening-up, Chinese cities are gradually shifting from incremental expansion to stock development. As a megacity with a large population and little land, Shenzhen is one of the cities with the highest density of villages in China. As a special product of Shenzhen's rapid urban development, which has not kept pace with the urbanization process, most of the urbanized villages have common problems such as poor living quality, serious fire hazards, high security risks, insufficient public space and limited plant greening. This makes the village in the city the focus and difficulty of Shenzhen's urban renewal. At the same time, the urbanized village in Shenzhen is a silent recorder of the development context of the city and an important space carrier of the city's history and culture. It is reported that the purple line and historical landscape area of urban villages account for 41% of the historical and cultural space of the city. Most of the intangible cultural heritage above the municipal level originates from urbanized villages [7]. Their residential value and historical and cultural connotation have been concerned and recognized, which has changed the method of the renewal and reconstruction of urbanized villages in Shenzhen from the early demolition and reconstruction to comprehensive renovation. According to the Shenzhen Urban Master Plan (2010–2020) issued in 2010, the renewal of urbanized villages will focus on comprehensive renovation. In the Overall Plan for Comprehensive Improvement of Urbanized Villages (Old Villages) in Shenzhen (2019–2025) issued in 2019, it was pointed out that the average area of comprehensive improvement zones is 56%, and the central urban area is as high as 75%, so large-scale demolition and construction are not allowed in this area. Shenzhen was the first city to undergo reconstruction instead of the old, mainstream method of demolition/reconstruction. By restoring the original village buildings and directly converting urban villages into rental apartments after subleasing them to the developer, the gap between the lack of public housing and the existing supply becomes filled [8]. At the same time, the relationship between the main bodies in Shenzhen's urban renewal governance is also constantly transforming and upgrading. In the early stage of exploration, urban renewal was dominated by the government, and the power subject was in a passive position due to the lack of a voice. The government obtained development and construction land from the original power subject at a relatively low cost. With the continuous expansion of the scale of renewal, the government-led model has a difficult time bearing the high cost of renewal. At the same time, the drawbacks and problems of the previous top-down mandatory management are constantly exposed. The government has begun to allow the market to intervene and gradually formed a multi-agent coordination mechanism of a government market power subject. In recent years, humanism and harmony have become the key words of high-quality urban development. Shenzhen is paying more attention to the interests of diverse groups in urban renewal and governance. However, urban renewal can successfully help maintain the function and vibrancy of urban centers, and urban renewal can also lead to negative issues, including social exclusion and the discontinuity of social lives [9]. These issues are not only common in a Northern and Western context, but they also appear in the Southern sphere, including South Africa [10] and the Middle East [11].

Previous studies have proposed a set of critical factors and corresponding indicators to be considered when undertaking sustainable urban renewal [12]. The heritage conservation in urban renewal districts should not only address the physical fabric of the historic buildings and the surrounding environment but also the social impact and the intangible values of a community as a whole, as they are of paramount importance [13]. However, little work has been carried out that measures the post-renewal evaluation of urbanized villages with cultural resource, and a more systematic evaluation system based on multi

public satisfaction that promotes the sustainable renewal of older districts with cultural heritage is needed.

Nantou Ancient City in Shenzhen has been chosen as a case study since it is an urbanized village with rich historical and cultural resources. Since 2011, it has experienced urban renovation launched by the government and the developer. Recently, it has been transformed from an older residential district into complex districts, including the residence, commercial and travel functions. From the perspective of the public, this study explores the existing problems of renewal and transformation by establishing an evaluation system suitable for the characteristics of Nantou Ancient City. First of all, on the basis of combing the existing community renewal evaluation system, especially the satisfaction evaluation and the research on the historical value of urbanized villages, the historical and cultural value and the renewal and transformation process of Nantou Ancient City are introduced, and the multiple values of Nantou Ancient City are clarified. Secondly, the demand contradiction between the public group and the core stakeholders, as well as the internal demand and the difference of the public group, are analyzed. After that, combined with field research, literature analysis and network review data analysis, the post-renovation evaluation system of Nantou Ancient City is constructed in terms of six aspects: residential environment, supporting facilities, street space, history and culture, economy, commerce and social culture. Then, a questionnaire is designed for three different groups of residents, merchants and tourists, and the evaluation system is applied to Nantou Ancient City for empirical research. By means of mean analysis, variance analysis and IPA analysis, the similarities and differences of the evaluation of different public groups are compared, which is taken as the basis for summarizing the existing problems in the renovation of Nantou Ancient City, and optimization suggestions are put forward.

Nantou Ancient City contains historical and cultural values that need to be excavated under the style of an urbanized village, which is a special and precious sample of urban culture. In the existing research, the evaluation system for urbanized villages or historical blocks has a difficult time reflecting the particularity of Nantou Ancient City. This research comprehensively analyzes the value of Nantou Ancient City and the needs of the public, establishes a renewal evaluation system suitable for Nantou Ancient City and provides a demonstration of the organic renewal of urbanized villages in Shenzhen, which is still in the exploration stage. By building an appropriate evaluation system and introducing the public as the evaluation subject, we can deeply explore the existing problems after the renewal, summarize the experience and lessons and provide a basis for the sustainable development of Nantou Ancient City and the organic renewal of urbanized villages in Shenzhen.

## 2. Literature Review

### 2.1. Research on Satisfaction Evaluation in Post-Renewal Evaluation

Satisfaction evaluation in post-renewal evaluation is part of the Post Occupancy Evaluation (POE) of community renewal. For example, Mehdipanah evaluates the perceived impact of urban renewal on the well-being of residents. From the two dimensions of physical environment and social economy, indicators such as square renovation and residential building renovation were selected to evaluate the two Barcelona communities, and the negative and positive views of different groups on community renewal were obtained [14].

At present, the post-renewal evaluation of urbanized villages is mainly divided into two categories, including the evaluation of the renovation method and the evaluation of the implementation effect and the performance of the renovation. The renewal and transformation method is mainly based on the qualitative evaluation. As for the research on the implementation effect and performance evaluation of renovation and reconstruction, which is associated with satisfaction evaluation, Xu built a post-renewal evaluation system for urbanized villages and evaluated three typical urbanized village renovation projects in Hangzhou through expert scoring and objective data analysis [15]. Wang believes that the current renovation of urbanized villages is more of a manifestation of the will of the government and developers. As one of the main stakeholders of the renovation, the rights

and interests of the villagers have not been fully guaranteed. Then, from the perspective of villagers, a systematic evaluation of 24 urbanized villages in Zhengzhou was carried out in terms of four aspects: material space, society, economy and environment after the renovation [16].

The satisfaction evaluation in post-renewal evaluation defined in this study is based on the POE, combined with the effect of the update and the status of the block after the update, with architecture and the environment as the main dimensions, and at the same time considering the aspects of history and culture, block commerce, social and humanities, etc., to build a complete evaluation system and make a more comprehensive evaluation. This study evaluates the updated environment and its impact on the basis of public needs and its value orientation and focuses on public satisfaction after the update.

*2.2. Research on the Post-Renewal Evaluation of Historic Districts*

The evaluation of the renewal of historic districts mostly focuses on evaluating the social impact brought by the renewal. Yung pointed out that the renewal of historical districts should not only solve the problems of historical buildings and their surrounding physical environment but also consider the social impact on the whole community [13]. The renewal of historical districts has gradually transformed from demolition to protection and revival, and there are still social problems that cannot be ignored [17]. During the renewal process, the forced eviction of indigenous peoples and original business is a common phenomenon. In addition, urban renewal often prioritizes economic growth and material improvement, which may lead to the homogenization tendency of the renewal area and thus the lack of place identity [18]. For example, Rui investigated the actual feelings of local residents through questionnaires and found that the respondents in the Longsheng County Landscape Reconstruction Project paid more attention to the impact of the regional culture of intangible space on the urban landscape [19].

In the conservation and renewal of urban historic districts, social inclusion is a key element to achieving conservation and renewal. Social inclusion recognizes the diversity of social cultures and groups and is committed to safeguarding the right of everyone to participate in decision making and promoting the fair distribution of urban space resources [20]. In the protection of historical districts, more and more attention is paid to fairness, the interests of social groups other than stakeholders are increasingly valued and space production is questioned by the academic circles as a means of capital accumulation. Zhang compared the differences in the spatial perception of residents and tourists and analyzed the reasons. He pointed out that the needs of "users" should be paid attention to in the reconstruction and evaluation of historical blocks [21]. Taking the Nanluoguxiang area as an example, Zhang studied the key factors in the urban renewal project of the historic district and the influencing relationship of each factor and used DEMATEL (Decision-making Trial and Evaluation Laboratory) to analyze the influence of practitioners, local officials, participating residents and merchants on each factor. According to the perception of factors, it was found that there are obvious differences in participants' perception of the commercialization of urban renewal projects [22]. The essence of the renewal of the historic district is to redistribute the land premium on the basis of coordinating the interests and demands of the government, the market and residents so as to achieve Pareto optimality [23]. Existing research and practice show that the participants in the renewal historic districts are currently concentrated on the core stakeholders, and the public awareness of participation in the renewal process is not high and there are few channels for participation. Thus, many researchers [24,25] agree that the renewal of historic districts should strengthen the power to ensure public participation.

In conclusion, in recent years, the trend of research on the renewal evaluation of historical districts has shifted from the evaluation of authenticity to the evaluation of the process of the renewal and protection of historical and cultural districts and the effect of renewal and protection. In the selection of evaluation subjects, the existing research focuses on tourists and local residents. The evaluation content includes physical space, block value,

historical and cultural characteristics, functional formats, etc., which provides a certain degree of foundation for the selection of evaluation indicators in our study. The status and role of the public group as a multi-subject in the renewal and renovation has been confirmed, but there are few studies on the multi public as the evaluation subject in the current urbanized village renewal evaluation.

### 2.3. Post-Renewal Evaluation Using the IPA Evaluation Method

Importance–Performance Analysis (IPA), introduced by Martilla and James [26], which uses two dimensions, "importance" and "performance", to prioritize the actions or improvements, is simple yet useful and is considered to have been widely adopted in research fields such as airports and airline services [27], tourism [28,29], hotels [30,31], etc. [32,33].

In the IPA method, by obtaining the mean value of importance and satisfaction, an IP map is made, and then four quadrants are located on the map according to the importance and satisfaction of each indicator. The IPA method reflects the gap between the actual and the ideal intuitively, so it is easy to be understood. Indicators in the first quadrant mean that importance and satisfaction are both high, while the second quadrant shows high satisfaction and low importance indicators, which need to be moderately regulated. The third quadrant is an active expansion area with low importance and satisfaction, and the fourth quadrant shows high importance but low satisfaction indicators, which need to be optimized urgently [34]. Combining importance and satisfaction for analysis can improve the interpretation of indicators and help in putting forward the targeted recommendations [35].

In related studies, IPA has become an important method for research on satisfaction. For example, the IPA scheme has been used as a tool to explore tourist satisfaction with 16 attributes of popular nature-focused boat tours [36]. Cai et al. attempted to evaluate the customer satisfaction of Japanese homestays for built-environment efficiency after the outbreak of the COVID-19 pandemic [37]. At the same time, online data and the IPA method have been applied in combination in recent years, such as Chen et al. introducing a novel approach for assessing tourist satisfaction by identifying attributes of interest from social media text [38].

In this study, IPA analysis is used to facilitate the diagnosis and decision making, and the data are classified and displayed in the form of a quadrant diagram so as to intuitively reflect various indicators. The evaluation results reflect the existing problems of the current renovation more clearly and provide a basis for the proposal.

### 3. Methodology

From the perspective of the multi public, this study takes the renovation of Nantou Ancient City as an example, and based on the existing relevant research, it explores and constructs a suitable evaluation system for Nantou Ancient City renovation and then summarizes the existing achievements and shortcomings of the renovation and proposes that it is feasible to implement follow-up optimization suggestions.

### 3.1. Construction of a Post-Renewal Evaluation System of Nantou Ancient City

The process of the post-renewal evaluation of Nantou Ancient City includes three parts: the initial construction of the evaluation system, the formal construction of the evaluation system and the analysis of the evaluation results (Figure 1). In the initial steps of constructing the evaluation system, first, the analysis of previous literature and online comment data, combined with the characteristics of Nantou Ancient City, constructed a preset index set. Second, a questionnaire survey was conducted among residents, tourists, merchants and experts, and an evaluation system was formally established after screening and determining a formal indicator set. Finally, the questionnaire was designed to collect data, and the commonality and differences of different public groups were analyzed to obtain the final evaluation results.

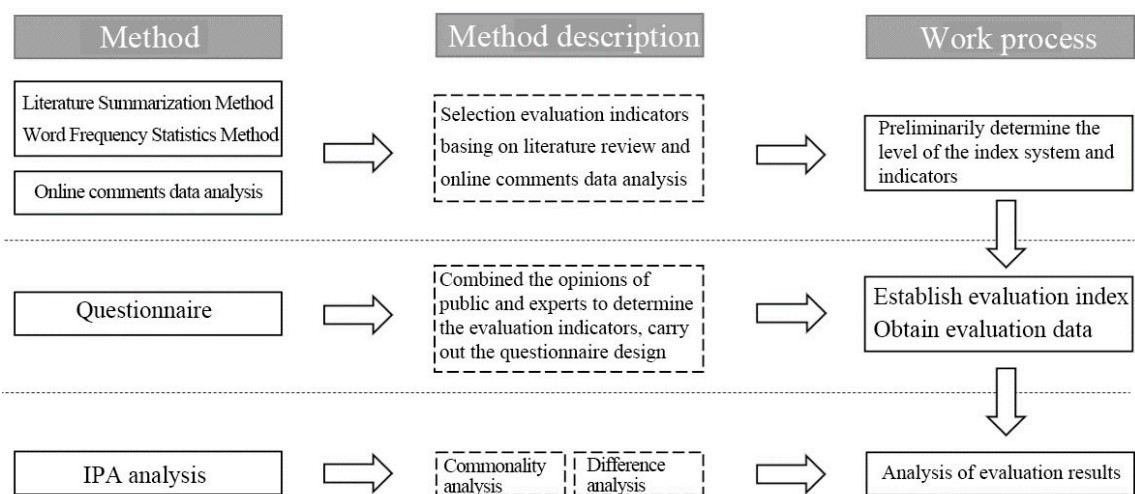

**Figure 1.** Evaluation process and method.

*3.2. Selection of Evaluation Indicators*

The selection of evaluation indicators is mainly carried out in terms of three aspects. First, analyze the previous literature, refer to the evaluation system and indicators of previous research, determine the first-level evaluation indicators and collect the second-level indicators that can be referenced. Second, analyze the online comment data of Nantou Ancient City, through text analysis of the comment content and the extraction of high-frequency words, to obtain the main concerns of visitors to Nantou Ancient City and supplement the secondary evaluation indicators. Finally, combined with the opinions of the public and experts, appropriate additions and deletions of evaluation indicators make the evaluation system not only reflect the needs and concerns of the public but also feasible.

Based on the literature analysis, the first-level indicators of the evaluation system in this study are determined as six items of living environment, public facilities, environment, history and culture, commercial economy and social humanities. The selection of secondary indicators refers to the frequency of indicators in the existing literature and also takes into account the actual situation of the renovation of Nantou Ancient City (Table 1). In addition, because the literature has its own research pertinence and the amount is limited, the final establishment of the second-level index needs to further consider the characteristics of Nantou Ancient City.

**Table 1.** Preliminary preset evaluation index set for the post-renewal evaluation of Nantou Ancient City based on literature analysis.

| First-Level Evaluation Indicator | Second-Level Evaluation Indicator |
| --- | --- |
| Living environment | Living area |
| | Living conditions (lighting, ventilation) |
| | Residential building quality |
| | Community security management |
| | Residence type |
| Public facilities | Number and distribution of rest and recreation facilities |
| | Number and distribution of toilets |
| | Accessibility to external transportation (bus, subway) |
| | Internal road accessibility (signs, guide) |
| Environment | Green landscape diversity |
| | Environmental hygiene quality |
| | Layered richness of public space |
| | The aesthetic feeling of street space |

**Table 1.** *Cont.*

| First-Level Evaluation Indicator | Second-Level Evaluation Indicator |
|---|---|
| History and culture | Harmony of old and new architectural styles |
| | Cultural display |
| Commercial economy | Economic benefits of the neighborhood |
| | Price reasonableness |
| | Format richness |
| Social humanities | Neighborhood relations |
| | Community belonging |
| | Public participation |

### 3.3. Analysis of Online Comments

Online comments are characterized by a large amount of review data and diverse reviewers. Reviewers include tourists, neighbor residents, citizens, etc. Existing research has confirmed that the review data have a reliable reference value for the evaluation of old city renovation and renewal [39]. With the help of web crawler technology, we obtain comment data about Nantou Ancient City from four websites, namely, Dianping (accessed on 2 June 2021), Ctrip.com (accessed on 7 June 2021), Qunar.com (accessed on 14 June 2021) and Mafengwo (accessed on 21 June 2021), and accumulatively obtain 2212 comment data. The top 100 high-frequency words are selected for sorting, mainly including five categories, namely, history and culture, commercial economy, public facilities, environment and social humanities. On this basis, the specific evaluation sentences are analyzed, refined and integrated with the evaluation system obtained in the previous literature, and the secondary indicators of the evaluation system are supplemented. Finally, 26 indicators such as living area and living conditions are selected as the second-level indicators. A set of preset evaluation indicators for the post-renewal evaluation of Nantou Ancient City.

### 3.4. Determination of the Final Evaluation System

By issuing a preliminary questionnaire for indicator selection, a public opinion survey was conducted for residents, tourists and merchants, which provided a reference for the selection of evaluation indicators and the optimization of the questionnaire form after the renewal of Nantou Ancient City. The set of pre-set evaluation indicators for the updated evaluation of Nantou Ancient City that was revised after the public opinion survey were re-tabulated, experts in relevant fields were consulted through the form of questionnaires, and the evaluation indicators were finally screened and determined to obtain the final evaluation system (Table 2).

**Table 2.** The final evaluation index set of the post-renewal evaluation of Nantou Ancient City.

| First-Level Evaluation Indicator | Second-Level Evaluation Indicator |
|---|---|
| A Residential environment | A1 Living conditions (lighting, air circulation) |
| | A2 Residential building quality (structure) |
| | A3 Community security management |
| B Supporting facilities | B1 Number and distribution of recreational facilities |
| | B2 Number and distribution of toilets |
| | B3 External traffic accessibility (transit, metro) |
| | B4 Internal road accessibility (identification, signature) |
| | B5 Number and distribution of parking lots |
| | B6 Quality of night lighting |
| C Landscape environment | C1 Diversity of greening landscape |
| | C2 Aesthetic feeling of the spatial form of the main street |
| | C3 The richness of public space |
| | C4 Quality of environmental sanitation |
| | C5 Shop decoration aesthetics |

**Table 2.** *Cont.*

| First-Level Evaluation Indicator | Second-Level Evaluation Indicator |
|---|---|
| D Historical culture | D1 Harmony of architectural style |
| | D2 Display of Shenzhen original culture |
| E Economy of the historic district | E1 Change in rent |
| | E2 Earnings |
| | E3 Rationality of consumer price |
| | E4 Abundance of business types |
| F Social humanities | F1 Neighborhood relations |
| | F2 Sense of community belonging |
| | F3 Public participation |
| | F4 Diversity of cultural activities |
| | F5 Attractiveness of exhibition contents |

*3.5. Application of the Post-Renewal Evaluation System of Nantou Ancient City*

Based on the constructed evaluation system of Nantou Ancient City after the renewal, the questionnaires were designed and distributed according to the different needs and characteristics of residents, merchants and tourists.

(1) The form of the questionnaire

Considering that the study focuses on the perspective of the multi public, the evaluation subject selects residents, merchants and tourists and determines the distribution form of the final questionnaire according to the characteristics of multi public groups. Among them, the residents' questionnaires are distributed in the form of paper questionnaires; the merchants' questionnaires are in the form of paper questionnaires, avoiding distribution during the noon and evening business peaks; the tourist questionnaires are in the form of a combination of online questionnaires and paper questionnaires, mainly online.

(2) Contents of the questionnaire

The formal questionnaire consists of two parts: satisfaction–importance evaluation and the basic information of respondents. The satisfaction and importance evaluation part is scored according to the Nantou Ancient City post-renewal evaluation system constructed above, and the Likert scale is used to divide the evaluation into five measurement levels and respectively assign values to them.

**4. Case Study**

*4.1. Overview of Nantou Ancient City*

Nantou Ancient City is a testimony to the early urban development of Shenzhen. In the history of Shenzhen, the county government was established in Nantou Ancient City. The history of Nantou Ancient City can be traced back to the Three Kingdoms period. The history of the Nantou Ancient City area is more than 1700 years old, and its construction history is more than 600 years old (Figure 2). It is located in the north of the Nantou overpass, Nanshan District, Shenzhen, Guangdong Province. The widest point from east to west is about 460 m, and the longest point from north to south is about 420 m. The total area is about 14.5 hectares. Nantou Ancient City basically follows the pattern of "Six Verticals and One Horizontal" in the Ming Dynasty. There are more than 1000 buildings inside, and the building density is as high as 50%. Most of the buildings are modern residential buildings in urbanized villages, accommodating more than 30,000 residents.

The research scope of the post-renewal evaluation is selected as the entire ancient city living area, including the renovated area and the unrenovated urbanized village residential area, as shown by the red line in the figure (Figure 3). The renovation scope of Nantou Ancient City in 2019 is shown by the yellow line in the figure, including the main cross street and part of the southwest area. Although the physical space change of this renovation has spatial limitations, considering that the social, economic and other aspects of the renovation are not limited to the scope of renovation, the selection of evaluation indicators and the

distribution of questionnaires comprehensively considered the entire living area of the ancient city.

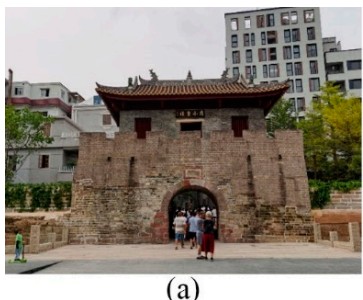
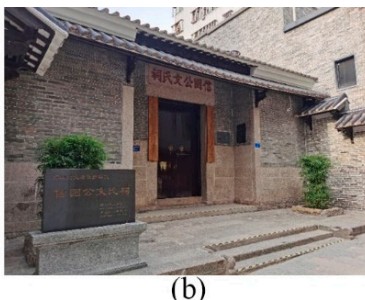
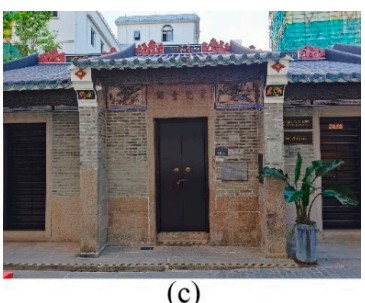

(a)  (b)  (c)

**Figure 2.** Primary historical buildings of Nantou Ancient City ((**a**). South Gate of Nantou Ancient City, (**b**). Xinguogong Wen Ancestral Hall, (**c**). Dongguan Club).

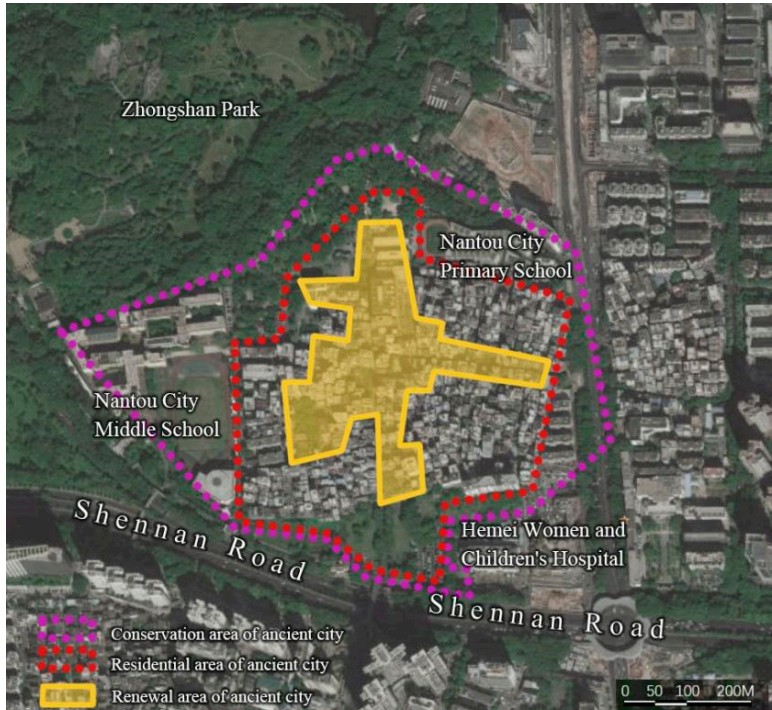

**Figure 3.** Study area.

*4.2. Research on Nantou Ancient City*

The research on Nantou Ancient City can be roughly divided by the event of the 2017 Shenzhen–Hong Kong Urban Biennale. Prior to the event, studies were mostly carried out on the historical evolution and value judgment of Nantou Ancient City, reviewing and sorting out the historical development context and cultural relics of Nantou Ancient City and analyzing the value of Nantou Ancient City. After the event, the academic circles continued to pay more attention to the renovation and reconstruction of Nantou Ancient City. The research perspective shifted from history to the present and the future and discussed the spatial status and future development of Nantou Ancient City. The School of Architecture of Tsinghua University launched an experimental course to explore the renovation strategy of Nantou Ancient City in terms of multiple dimensions of excavating the original sense of place and creating public space [40]. From the perspective of historical humanities and cultural revival, Meng proposed that the development history of Nantou Ancient City is the condensed history of Shenzhen's urban development, and the renewal and rebirth of the ancient city should focus on the improvement of residents' quality of

life and the promotion and revival of local culture [41]. From the perspective of multiple subjects, the renovation of Nantou Ancient City needs to take into account the opinions of all stakeholders, pay attention to the construction of a platform for the relationship between multiple publics and explore a renewal model that gives full play to the value of multiple publics [42]. Interactive participation contributes to stronger civic structures and more sustainable places [43].

*4.3. Analysis of the Demands and Contradictions of the Multi Public and Core Stakeholders*

In the process of renovation, the public, as a non-core stakeholder, has not been fully considered for its needs. At the same time, there is a certain conflict between public demand and the triangular pursuit of interests formed by the government–developer–village (Figure 4).

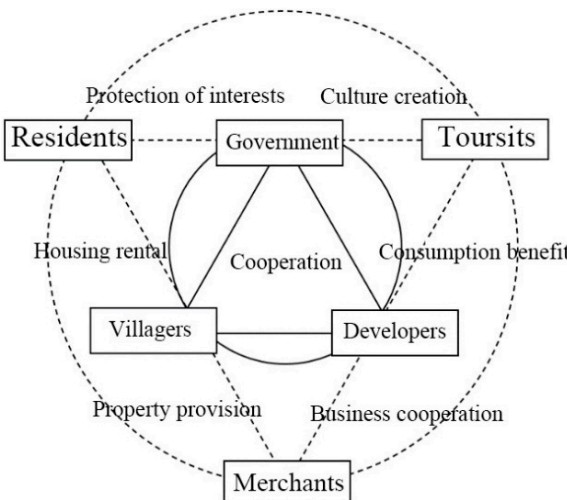

**Figure 4.** Diagram of the relationship between the public demand and core stakeholders.

For the residents, low-rent housing and a pleasant community environment are their main demands for the renewal environment. In terms of housing and rents, due to the complex interests and ownership relations involved in the reconstruction of urbanized villages, the government does not adequately protect the real interests of residents. After renovation, it is often accompanied by rising rents, population replacement, the large-scale relocation of tenants and the disruption of the original social network. In terms of space quality improvement, developers' investment in the improvement of public space quality often focuses more on meeting the needs of tourists so as to facilitate the operation and profitability of the block and pays insufficient attention to the daily life of local residents.

For the tourists, authentic cultural experiences and unique cultural consumption are the main appeals of their visit to the renewal area. In the pursuit of capital and interests, the government's deliberate pursuit of image projects and the excessive commercialization brought about by renovations have led to the homogenization of neighborhoods to a certain extent, the cultural experience is not so interesting, but the consumer prices remain high.

For the merchants, realizing the profitability of the shops is the main appeal for them to settle in the newer areas. The profitability of a store mainly depends on the costs and benefits of daily operations. The biggest cost for merchants is rent. With the continuous deepening of neighborhood renovation, the impact of gentrification has become increasingly prominent, and regional rents are on the rise. It is easy to make it difficult for merchants to operate for a long time, who are eventually forced to move out.

The public's recognition of the renovation is an important factor in determining whether the renovation can be sustainable. The contradiction between the public and the core stakeholders is the key point and difficulty of the renovation, which should be paid attention to in the post-renewal evaluation.

## 5. Results

### 5.1. Questionnaires

The formal questionnaire was distributed four times, including the holidays of 13–14 June 2021 and the working days of 15 June and 16 June. A total of 150 questionnaires were distributed four times, including 50 for residents, 50 for merchants and 50 for tourists. Among the returned questionnaires, 129 are valid and 46 are valid, with an effective rate of 92%; there are 42 valid questionnaires from merchants, with an effective rate of 84%; there are 40 valid questionnaires for tourists, with an effective rate of 80%.

### 5.2. Reliability Analysis and Validity Test

This study used IBM SPSS Statistics 25 software (International Business Machines Corporation, Armonk, NY, USA)to analyze the reliability of the questionnaires of residents, merchants and tourists. The reliability of the satisfaction scale of each subject, the important scale and the total scale is greater than 0.8, indicating that the reliability of the questionnaire is good. This indicates that the opinions and attitudes measured by the questionnaire are highly reliable.

This study used IBM SPSS Statistics 25 software to analyze the validity of the questionnaires of residents, merchants and tourists, respectively. The KMO values of the validity of the satisfaction scale of each subject and the important scale are greater than 0.6, and the *p* values are less than 0.05. The validity of the questionnaire meets the requirements. This shows that the measurement results of the questionnaire are valid.

### 5.3. Statistics of Respondents' Basic Information

For the diversified public, statistics were collected on the gender, age, occupation, education background, place of origin, length of residence and family population of the interviewed residents; statistics were collected on the gender, age, education background and store type of the merchants interviewed; statistics were collected on the sex, age, occupation, education background, residence, number of visits and purpose of the visitors.

### 5.4. Analysis

Based on the analysis of the questionnaire data, the basic information of the respondents is summarized. On this basis, the respondents' basic understanding of Nantou Ancient City is investigated, and an IPA analysis and comparative analysis of each subject are conducted.

#### 5.4.1. Cognitive Analysis of the Positioning of Nantou Ancient City

Residents, merchants and tourists have formed a certain recognition of the history and culture of Nantou Ancient City. The basic cognition is that the urbanized villages with history and culture are the main ones. However, the response rates of building Nantou Ancient City into a cultural innovation industry district, a traditional urban style exhibition area and a humanistic tourism destination mentioned in the government's renovation goals are similar and low, indicating that there is still a certain distance to achieving this goal (Figure 5).

#### 5.4.2. Analysis of the Value Identification of Nantou Ancient City

Value recognition is the guide to the direction of renewal, and there are differences in the means of renewal under different values. The research was carried out by means of multiple topics in the questionnaire to analyze the value recognition of various public groups for Nantou Ancient City. Although the value orientations most recognized by various subjects are different, in general, the response rates of various values of Nantou Ancient City are at a similar level, which shows that the historical value, economic value and residential value of Nantou Ancient City are very important for the public groups. In the future transformation, the embodiment and balance of multiple values with humanistic tourism and the living quality of the residents should be paid attention to (Figure 6).

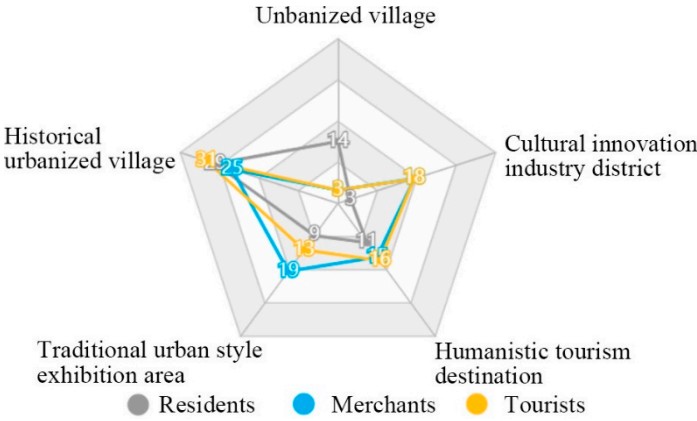

**Figure 5.** Cognitive analysis of the positioning of Nantou Ancient City by the multi public.

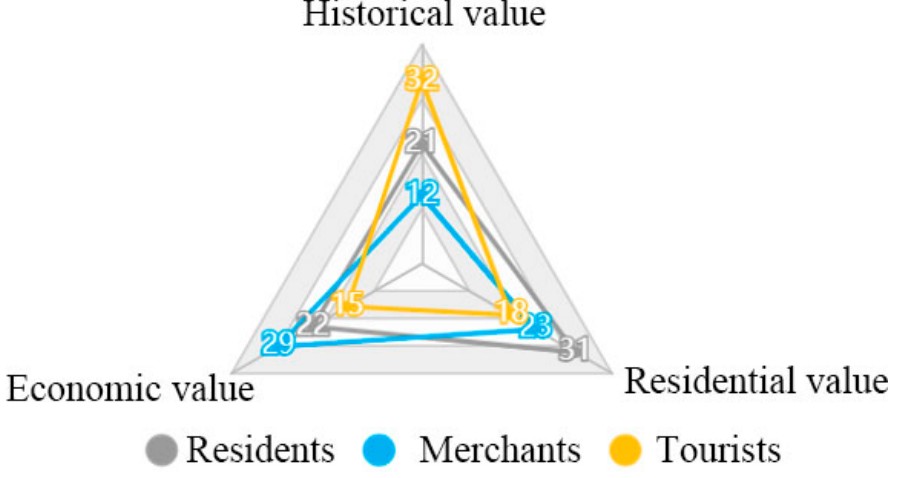

**Figure 6.** Analysis of the value identification of Nantou Ancient City by the multi public.

5.4.3. Comparative Analysis of the Overall Satisfaction of Each Subject

By calculating each index of a single sample, the overall satisfaction with the renewal and reconstruction of Nantou Ancient City is calculated. The results show that the satisfaction for different categories of samples shows a significant difference ($p < 0.05$), and the overall satisfaction of tourists is significantly higher than that of residents (Table 3).

**Table 3.** Variance Analysis of the Overall Satisfaction of All Subjects.

| | Type (Average Value $\pm$ Standard Deviation) | | | F | $p$ |
|---|---|---|---|---|---|
| | Resident (n = 46) | Merchant (n = 42) | Tourist (n = 40) | | |
| Satisfaction | $3.39 \pm 0.40$ | $3.56 \pm 0.45$ | $3.70 \pm 0.70$ | 3.669 | 0.028 * |

* $p < 0.05$.

5.4.4. Comparative IPA Analysis of Multiple Publics (Residents, Merchants, Tourists)

The importance is taken as the abscissa, and the mean value of the importance of all the secondary evaluation indicators is taken as the boundary value of the importance; satisfaction is the ordinate, and the average of the satisfaction with all secondary evaluation indicators is the boundary value of satisfaction, which builds a coordinate system, and then the IPA analysis charts of residents, merchants and tourists are attained (Figure 7).

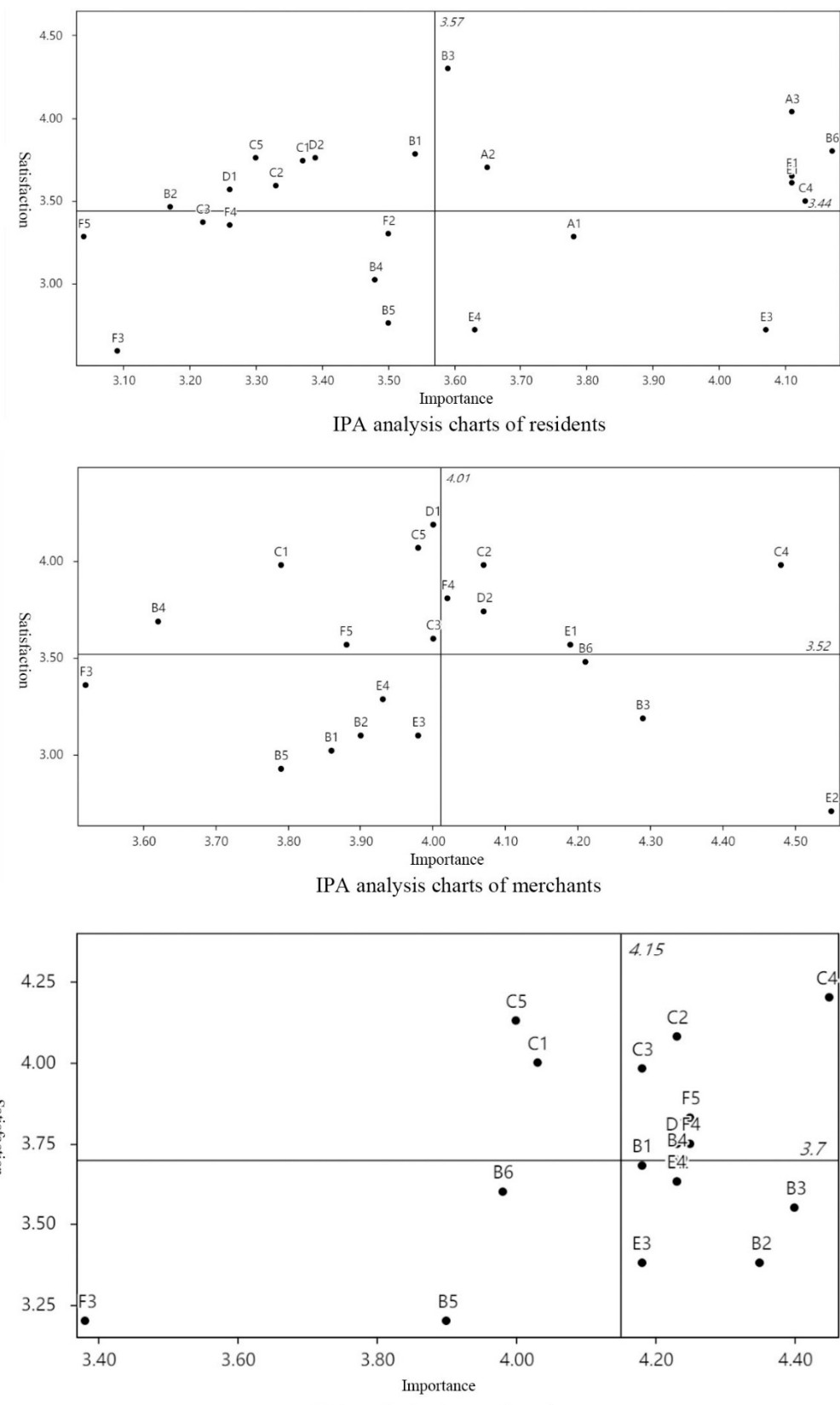

**Figure 7.** IPA analysis of the multi public.

By comparing the IPA quadrants of residents, businesses and tourists, we mainly focus on the existing advantages of quadrant I and the areas in quadrant IV that need to be improved so as to identify the current transformation effects and problems.

It can be seen from Table 4 that the common indicators of two or more groups in quadrant I include C2 the aesthetic feeling of the spatial form of the main street, C4 the quality of environmental sanitation, E1 the change in rent and F4 the diversity of cultural activities, which shows that the effects of the four indicators after transformation are recognized by most people. The indicators that are common to two or more groups in quadrant IV include B3 external traffic accessibility, E3 the rationality of consumer prices and E4 the abundance of business types, which shows that the effects of these three indicators after transformation have a difficult time meeting the needs of most people.

**Table 4.** Difference in IPA from the public.

| Evaluation Index of the First Level | Evaluation Index of the Second Level | Evaluation Result of Residents | Evaluation Result of Merchants | Evaluation Result of Tourists |
|---|---|---|---|---|
| A Residential environment | A1 Living conditions (lighting, air circulation) | IV | — | — |
| | A2 Residential building quality (structure) | I | — | — |
| | A3 Community security management | I | — | — |
| B Supporting facilities | B1 Number and distribution of recreational facilities | II | III | IV |
| | B2 Number and distribution of toilets | II | III | IV |
| | B3 External traffic accessibility (transit, metro) | I | IV | IV |
| | B4 Internal road accessibility (identification, signature) | III | II | I/IV |
| | B5 Number and distribution of parking lots | III | III | III |
| | B6 Quality of night lighting | I | IV | III |
| C Landscape environment | C1 Diversity of the greening landscape | II | II | II |
| | C2 Aesthetic feeling of the spatial form of the main street | II | I | I |
| | C3 The richness of the public space | III | II | I |
| | C4 Quality of environmental sanitation | I | I | I |
| | C5 Shop decoration aesthetics | II | II | II |
| D Historical culture | D1 Harmony of architectural style | II | II | I |
| | D2 Display of Shenzhen original culture | II | I | IV |
| E Economy of the historic district | E1 Change in rent | I | I | — |
| | E2 Earnings | — | IV | — |
| | E3 Rationality of consumer price | IV | III | IV |
| | E4 Abundance of business types | IV | III | IV |
| F Social humanities | F1 Neighborhood relations | I | — | — |
| | F2 Sense of community belonging | III | — | — |
| | F3 Public participation | III | III | III |
| | F4 Diversity of cultural activities | III | I | I |
| | F5 Attractiveness of exhibition contents | III | II | I |

## 6. Discussion

Based on the IPA analysis, the indicators in quadrant IV can be summarized. It can be seen that, based on the needs of different public groups, there are differences and common problems in Nantou Ancient City.

### 6.1. Common Problems and Analysis

#### 6.1.1. High Consumer Prices

Both residents and tourists believe that the current consumption price of Nantou Ancient City is on the high side. The rental price of Nantou Ancient City, which has risen sharply due to the transformation investment, is the root cause of the current rising consumer price.

#### 6.1.2. The Richness of Business Type Is Low

Both residents and tourists believe that the current commercial richness of Nantou Ancient City needs to be improved. First, the newly settled stores had an impact on the original life service businesses, and the imbalance between the new and old businesses led to the collapse of the original market smoke, fire and gas, reducing the convenience of residents' lives. Second, the cultural connotation of the newly settled cultural and entrepreneurial businesses had a low correlation with Nantou Ancient City, and the creative industries were wrapped with a strong commercial atmosphere, which had a difficult time forming the consumer identity of tourists (Figure 8).

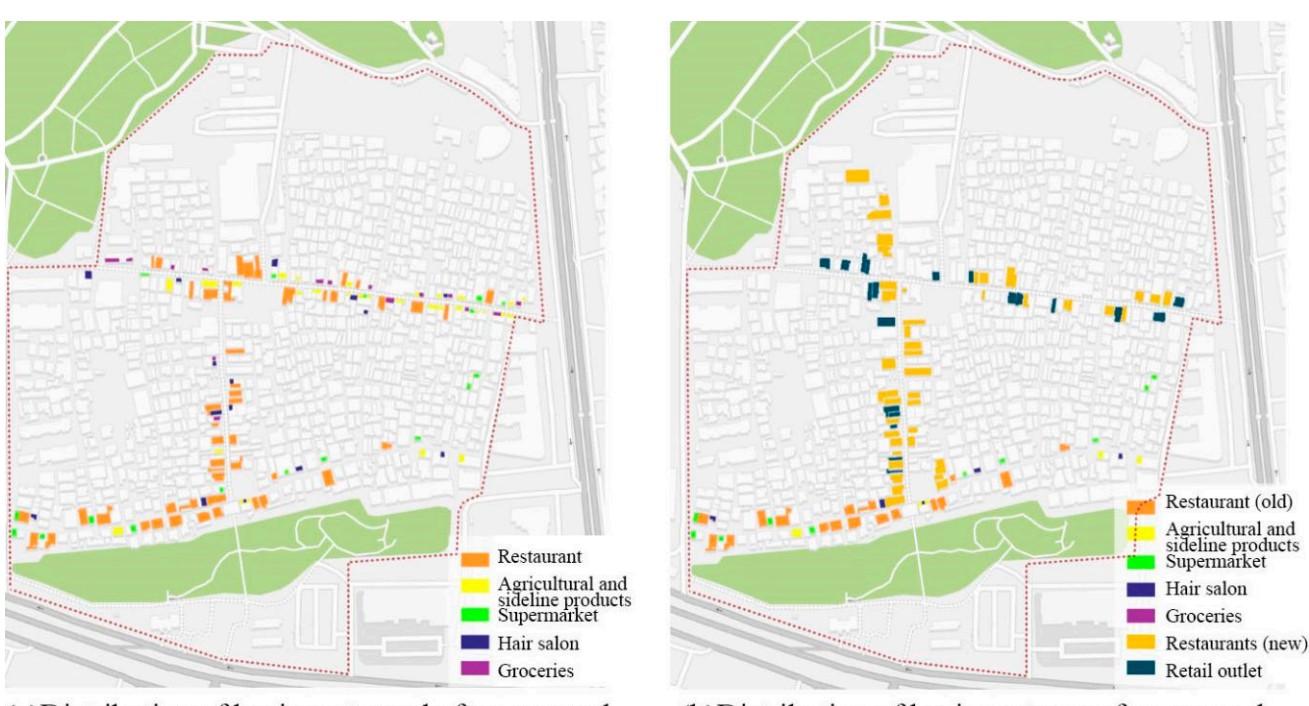

**Figure 8.** Comparison of business types before and after the renewal of Nantou Ancient City (Redrawn based on reference [44]).

#### 6.1.3. Low Accessibility of External Traffic

Both merchants and tourists believe that the current accessibility of external traffic in Nantou Ancient City is low (Figure 9).

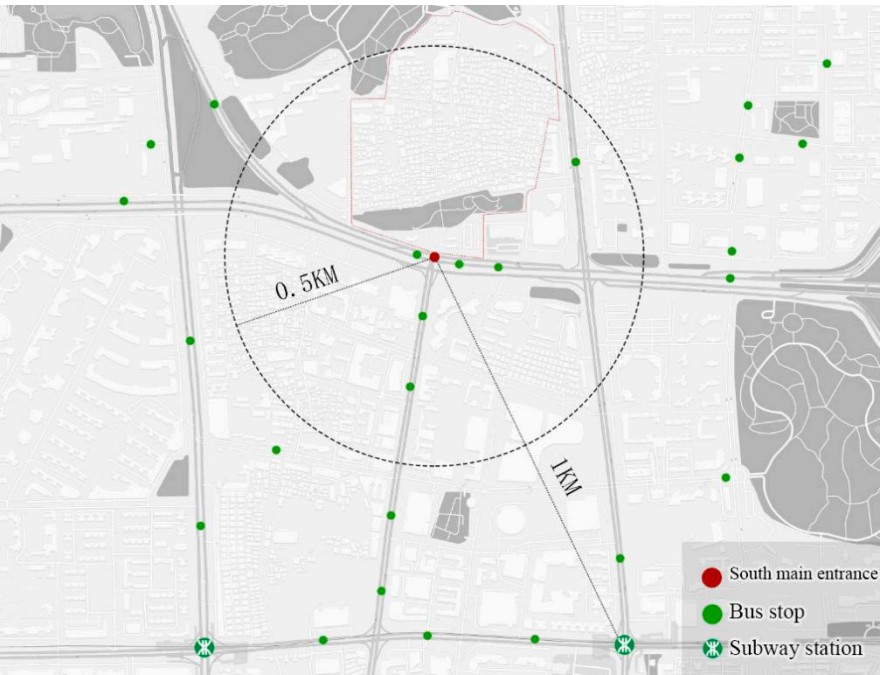

**Figure 9.** Exterior traffic situation of Nantou Ancient City.

*6.2. Differences and Analysis*

6.2.1. Residents: Living Conditions Need to Be Improved

The residential buildings in Nantou Ancient City have the common problems of urbanized villages. Owing to the property right of original villagers, the transformation of residential space is limited.

6.2.2. Merchants: The Basic Lighting at Night Is Insufficient, and the Income Is Not Ideal

For merchants, the improvement of basic lighting at night is more important than the landscape design of lighting. After reconstruction, the nighttime lighting of Nantou Ancient City has been redesigned, but the improvement of basic lighting is slightly insufficient. When the store is closed, the lighting level is low, and the local brightness of the street is easily low, resulting in an inconvenience for the merchants travelling after business. For the old merchants, most of their shops are catering and life services. The main audience of the shops is the residents of the ancient city. Since the transformation, the residents of Nantou Ancient City have lost a lot, which has affected the operation of the shops.

6.2.3. Tourists: The Number and Distribution of Recreational Facilities Are Unreasonable, Toilet Signs Are Weak and the Display of Shenzhen's Origin Culture Is Insufficient

The number and area of public spaces in Nantou Ancient City are small, and there are few places for the design of recreational facilities. At the same time, the overall distribution of recreational facilities has not been reasonably considered. The distribution of public toilets is relatively scattered, and the identification system is relatively weak, which makes it difficult to find toilets. Although the business types and activities settled and held after the transformation have enriched the cultural diversity of Nantou Ancient City, its theme is relatively new and modern, and it has a low degree of compatibility with the original culture of Nantou Ancient City.

*6.3. Suggestions on the Renewal and Optimization of Nantou Ancient City*

Based on the current situation of renewal summarized above, it can be seen that the current problems mainly focus on four dimensions, including economy and commerce, supporting facilities, living environment and history and culture. It puts forward optimization suggestions in terms of three aspects: focusing on the development of diversified business

types, improving the construction of supporting facilities and enhancing the display of history and culture.

### 6.3.1. Focusing on the Development of Diversified Business Types

(1) Diversified commercial types

The transformation replaced the original business types of the Cross Main Street with high-end catering and cultural innovation retail, which tended to be elitist and raised the overall consumption level of Nantou Ancient City. When attracting the investment, controlling the proportion of medium- and high-end businesses should be paid attention to in order to respond to the actual demands of local residents in daily life. The consumption orientation and acceptable price range of foreign tourists should be considered carefully so as to keep the consumption price of Nantou Ancient City in a reasonable range.

(2) Properly retaining the original life service stores

The pursuit of economic interests in transformation has forced the needs of residents to be transferred to tourists. However, tourists are only temporary visitors, and residents are the long-term residents. Especially with the repeated epidemic, tourists are limited in number, and residents are still the main activity groups and service objects of Nantou Ancient City. In the future reconstruction, attention should be paid to the preservation of life service stores, such as small supermarkets and barbershops, in order to maintain the balance between tourism development and daily life.

### 6.3.2. Improving the Construction of Supporting Facilities

(1) Additional recreational facilities

At present, there are few seats available for parking around the square. On the one hand, fixed seats can be set around the square along the existing flower beds, which not only ensures the relative privacy of the rest space but also does not affect the daily use of the square. On the other hand, mobile devices are set to provide visitors with a rest space when there is no exhibition activity in the square, and they can be moved to both sides of the square when holding activities so that the square space can be more fully used.

(2) Improving night lighting conditions

The nighttime lighting in Nantou Ancient City has a variety of types and forms. The public satisfaction of key lighting and performance lighting is high, but it is slightly insufficient for functional lighting. This is mainly reflected in the insufficient brightness of the functional lighting, especially the difficulty in meeting the needs of merchants for night travel. The functional lighting at night shall be designed according to the principles of safety and comfort. First of all, the location distribution of lighting facilities should be optimized, and reasonable spacing should be controlled to ensure the formation of a continuous linear space. At the same time, the number of lighting fixtures should be increased at space nodes with a large pedestrian flow. It is also necessary to consider the uniformity of the brightness of the lighting facilities to avoid glare caused by strong contrast, resulting in poor visual perception.

### 6.3.3. Enhancing the Display of History and Culture

Tourists responded that the current transformation did not fully display the history and culture of Shenzhen. Although the current weekend fairs, temporary exhibitions and other activities are rich, they are not connected with the history and culture of Nantou Ancient City. It is suggested to add historical and cultural education activities and plan the tour routes of cultural attractions.

(1) Adding historical and cultural education activities

At present, the linkage effect between the daily cultural activities of Nantou Ancient City and Nantou Ancient City Museum is weak. Many tourists do not visit Nantou Ancient City Museum when they visit the ancient city, or they cannot even understand that there are museums around the ancient city. It can be combined with the voluntary explanation of Nantou Ancient City Museum, from indoors to outdoors and from image to reality,

to form a more three-dimensional and rich tour flow line, to help visitors have a more comprehensive and intuitive understanding of the development history of Nantou Ancient City, to understand Shenzhen's origin culture, to form a weekly routine activity and to open the reservation entrance on the official WeChat account for online reservations.

(2) Designing tour routes for cultural attractions

The planning and design of cultural tour routes are focused on by connecting historical and cultural-related buildings such as Nantou Ancient City Museum, Nantou Exhibition Hall, Dongguan Guild Hall, Xin'an County Yamen and Baode Temple. Nodes are set on the route to introduce the historical stories of each development period. At the end of the route, a performance platform can be set to show videos in combination with the old functions of the Kole Square and to introduce the development process of the historical ancient city, villages in the city and cultural blocks of Nantou Ancient City. This enhances the visitors' overall understanding of the historical development and current status of Nantou Ancient City.

## 7. Conclusions

This paper investigates the multi public satisfaction key evaluation factors of urbanized villages with cultural resource. The study advocates that the evaluation system of older districts should address the interests of the multi public. Furthermore, this paper institutes the post-renewal evaluation system of urbanized villages, comprising 6 first-level indicators and 25 second-level indicators, which not only include the physical environment factors but also the social humanities and intangible values of older districts after renovation.

As for the renewal of Nantou Ancient City, the residents' satisfaction is the lowest. The reason for this is that the renewal is intended to make Nantou Ancient City a new cultural landmark in Shenzhen. Through the restoration of traditional styles, the introduction of a large number of cultural and creative shops and the planning and holding of festivals, it caters to the consumption and entertainment needs of tourists. At the same time, the policy of rents being free for the first year has been implemented for newly settled businesses to create a good business atmosphere. In contrast, the residential value and function of Nantou Ancient City have been ignored and weakened in the renewal, which makes it difficult to meet the needs of residents. Although the basic environment and public space have been improved, most of them are based on the needs of foreign tourists, such as adding road signs, toilets, etc.

The most concerning practical issues, such as living conditions, have not been solved in this reconstruction. The historical and cultural value of Nantou Ancient City cannot be ignored, and the excavation and display of the cultural context is something that both the government and the public enjoy. However, apart from the former historical city, Nantou Ancient City is now a living place for more than 30,000 residents. The cultural promotion of Nantou Ancient City should not be achieved at the expense of the convenience of residents' daily life, and the direction of renovation should also focus more on solving residents' practical demands.

**Author Contributions:** Conceptualization, S.G.; methodology, H.M.; software, S.G.; validation, J.L., investigation, M.W.; resources, S.G.; data curation, S.G.; writing—original draft preparation, S.G.; writing—review and editing, H.M.; J.L.; supervision, H.M.; project administration, H.M.; funding acquisition, H.M. All authors have read and agreed to the published version of the manuscript.

**Funding:** This research was funded by the National Social Science Foundation of China, grant number 20FGLB057.

**Data Availability Statement:** Not applicable.

**Acknowledgments:** The authors gratefully acknowledge that the funding for this research was provided by the National Social Science Foundation of China (Grant No. 20FGLB057).

**Conflicts of Interest:** The authors declare no conflict of interest.

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
