# Peer review of "Post-Renewal Evaluation of an Urbanized Village with Cultural Resources Based on Multi Public Satisfaction: A Case Study of Nantou Ancient City in Shenzhen"

_land, doi:10.3390/land12010211_

Round 1
Reviewer 1 Report
Overall, this is a very good paper. The subject of the manuscript, the methods used, the conclusions, and its impact on science on the article are all sound. The issue is that the subject of the article is so narrowly focused on Nantou Ancient City that the data and information provided are not relevant to anywhere else in the world. Why should you in Romania, or I in the United States, are more importantly our students in our respective countries who do not know any better, care about what is happening in the urban renewal of Nantou Ancient City? I don't know about you, but neither I nor my students have ever been nor have any understanding on what it is like there. Since this is a journal with an international audience, the authors need to be able to explain to everyone else who would read it what they can learn from their case study as well as how the lessons from this case study are relevant to people and communities living in places like Romania, the United States, as well as Africa, or South America. Explaining why a specific case study of one little place in the world is relevant to everyone in the world is important for the sake of informational relevance. This is explaining to the reader what is called the "Particular to the General." Or in other words explaining, why is the urban renewal of Nantou Ancient City important to know for everyone else in the world. I am sure that the urban renewal of Nantou Ancient City is very important to the people that live there, but please explain why this is important to those who doesn't live in Nantou. Otherwise, the journal creates the problem where it has published case studies on a plethora of places from all over the world, but no can explain their meaning behind them to anyone else.
Author Response
Please see the attachment, which include the manuscript after modification. Thank you so much.
Response to Reviewer 1 Comments
Reviewer 1: Overall, this is a very good paper. The subject of the manuscript, the methods used, the conclusions, and its impact on science on the article are all sound. The issue is that the subject of the article is so narrowly focused on Nantou Ancient City that the data and information provided are not relevant to anywhere else in the world. Why should you in Romania, or I in the United States, are more importantly our students in our respective countries who do not know any better, care about what is happening in the urban renewal of Nantou Ancient City? I don't know about you, but neither I nor my students have ever been nor have any understanding on what it is like there. Since this is a journal with an international audience, the authors need to be able to explain to everyone else who would read it what they can learn from their case study as well as how the lessons from this case study are relevant to people and communities living in places like Romania, the United States, as well as Africa, or South America. Explaining why a specific case study of one little place in the world is relevant to everyone in the world is important for the sake of informational relevance. This is explaining to the reader what is called the "Particular to the General." Or in other words explaining, why is the urban renewal of Nantou Ancient City important to know for everyone else in the world. I am sure that the urban renewal of Nantou Ancient City is very important to the people that live there, but please explain why this is important to those who doesn't live in Nantou.Otherwise, the journal creates the problem where it has published case studies on a plethora of places from all over the world, but no can explain their meaning behind them to anyone else.
Response 1: Please provide your response for Reviewer 1. (in red)
Thank you very much for the valuable suggestions of the reviewer. On the part of Introduction, the meaning of “particular to the general” has been added up from Line 103 to Line 117, including “Previous studies have proposed a set of critical factors and corresponding indicators to be considered when undertaking sustainable urban renewal (Lee,2003; Chan & Lee,2007). The heritage conservation in urban renewal districts should not only address the physical fabric of the historic buildings and the surrounding environment, but also the social impact and the intangible values of a community as a whole as they are of paramount importance (Esther et al., 2017). However, little work has been done that measures the post renewal evaluation of urbanized villages with cultural resource, and a more systematic evaluation system based on multi public satisfaction that promote the sustainable renewal of older districts with cultural heritage is needed.”
The reasons of Nantou Ancient City selected as the case study has been added from Line 118 to Line 122.
Moreover, the abstract of the article has added up the meaning of “particular to the general” from Line 12 to Line 15.

Reviewer 2 Report
I think that this article is very interesting and is worth being published in the Land journal. However, before the final publication, I would suggest making several revisions:
1 - I think the title is too long. I would suggest choosing a more concise title.
2 - The Methodology section is a bit too short. I would suggest demonstrating in greater detail your research approach. This section should explain how you will conduct research later on in your paper. Try to be less descriptive. It would be useful to make this section as concrete as possible.
3 - The Conclusion section is too short and therefore should be expanded. Here you should explain explicitly the contributions of your research project to the respective academic field.
Author Response
Please see the attachment, which include the manuscript after modification. Thank you so much.
Response to Reviewer 2 Comments
Point 1: I think that this article is very interesting and is worth being published in the Land journal. However, before the final publication, I would suggest making several revisions:
1 - I think the title is too long. I would suggest choosing a more concise title.
Response 1: Please provide your response for Reviewer 2. (in red)
Thank you very much for the valuable suggestions of the reviewer. The title is shorten into “Post Renewal Evaluation of Urbanized Village with Cultural Resource Based on Multi public Satisfaction: A Case Study of Nantou Ancient City in Shenzhen”.
Point 2: The Methodology section is a bit too short. I would suggest demonstrating in greater detail your research approach. This section should explain how you will conduct research later on in your paper. Try to be less descriptive. It would be useful to make this section as concrete as possible.
Response 2:
Thank you very much for the valuable suggestions of the reviewer. Some contents of Result section have been moved into the part of Methodology section, which including 3.1. Construction of post renewal evaluation system of Nantou Ancient City; 3.2. Selection of evaluation indicators; 3.3. Analysis of online comments; 3.4. Determination of the final evaluation system; 3.5. Application of the post renewal evaluation system of Nantou Ancient City, in order to explain how the research is conducted later in the paper. Please check the content from Line 265 to Line 343.
Point 3 : The Conclusion section is too short and therefore should be expanded. Here you should explain explicitly the contributions of your research project to the respective academic field.
Response 3:
Thank you very much for the valuable suggestions of the reviewer. The Conclusion section has added up the content of the contribution of the research on the respective academic field, please check the content from Line 636 to Line 642.
